# Association of Cord Blood Metabolic Biomarkers (Leptin, Adiponectin, IGF-1) with Fetal Adiposity Across Gestation [note 1]

**DOI:** 10.3390/ijms26146926

**Published:** 2025-07-18

**Authors:** Junko Tamai, Satoru Ikenoue, Keisuke Akita, Keita Hasegawa, Toshimitsu Otani, Marie Fukutake, Yoshifumi Kasuga, Mamoru Tanaka

**Affiliations:** Department of Obstetrics and Gynecology, Keio University School of Medicine, 35 Shinanomachi, Shinjuku-ku, Tokyo 1608582, Japan

**Keywords:** leptin, adiponectin, insulin-like growth factor-1, C-peptide immunoreactivity, fetal ultrasonography, fetal adiposity

## Abstract

Childhood obesity is a substantial health problem worldwide. The origin of obesity (increased adiposity) can be partly traced back to intrauterine life. However, the determinants of fetal fat deposition remain unclear. This study investigated the association between cord blood adipocytokines related to lipid metabolism (leptin, adiponectin, and insulin-like growth factor-1 [IGF-1]) and fetal adiposity during gestation. A prospective study was conducted in a cohort of 94 singleton pregnancies. Fetal ultrasonography was performed at 24, 30, and 36 weeks of gestation. Estimated fetal adiposity (EFA) was calculated by integrating measurements of cross-sectional arm and thigh fat area percentages and anterior abdominal wall thickness. Plasma cytokine levels and C-peptide immunoreactivity (as a proxy for fetal insulin resistance) were evaluated in cord blood samples obtained at delivery. The associations of cord blood leptin, adiponectin and IGF-1 levels with EFA at 24, 30, and 36 weeks were determined by multiple linear regression, adjusted for potential covariates. The multivariate analyses indicated that leptin was significantly correlated with EFA at 30 and 36 weeks. Leptin was also positively correlated with C-peptide immunoreactivity in the umbilical cord. Cord adiponectin levels were not associated with EFA across gestation. Cord IGF-1 levels were significantly correlated with EFA and estimated fetal body weight (EFW) at 36 weeks. In conclusion, cord leptin was associated with EFA at 30 and 36 weeks, and IGF-1 was associated with EFA at 36 and EFW at 36 weeks. In Conclusion, cord leptin was associated with EFA at 30 and 36 weeks, and IGF-1 was associated with EFA and EFW at 36 weeks. Considering the effects of leptin and IGF-1 on fetal insulin resistance and lipid metabolism, increased levels of leptin and IGF-1 are potential plasma biomarkers of increased fetal adiposity, which may predispose to infant obesity and metabolic dysfunction in later life.

## 1. Introduction

Childhood obesity and early-onset metabolic syndromes are substantial health problems worldwide, particularly in developed countries [1]. An estimated 35 million children under the age of 5 years were overweight in 2024. While just 2% of children and adolescents aged 5–19 were obese in 1990 (31 million young people), by 2022, 8% of children and adolescents were living with obesity (160 million young people) [2]. Recent epidemiological, clinical, and experimental investigations have revealed that the origins of obesity and adiposity can, in part, be traced back to fetal life [3,4]. Fetal adiposity is a predictor of newborn adiposity [5], which is associated with childhood obesity and future risk of metabolic dysfunction [3,4]. Hence, for the primary prevention of childhood obesity and early-onset metabolic syndrome, it is crucial to elucidate the antecedent conditions that affect fetal growth and body composition, especially fat mass.

Maternal pre-gravid weight had strong correlation with estimates of neonatal fat and percentage of body fat [6]. The accumulation of fetal fat can be explained by metabolic programming that can be influenced by maternal gestational weight gain [7,8]. Maternal plasma glucose levels and insulin resistance are well-known factors affecting fetal adiposity [5], newborn percentage body fat [9], and childhood obesity [10,11]. Several hormones and cytokines interact with glucose and insulin. Leptin is an adipokine that is synthesized in adipose tissue, and leptin concentration is associated with fat mass volume [12,13,14]. Leptin, through the induction of systemic inflammation, plays an important role in energy regulation, endocrine function (lipid metabolism and insulin resistance), immune response, and reproduction [13,15]. Adiponectin, adipocyte-derived protein that comprises 244 amino acids [16], correlates inversely with obesity [17]. Cord blood adiponectin levels are also related to insulin resistance and are decreased in gestational diabetes [18,19]. Insulin-like growth factor-1 (IGF-1) is synthesized especially in the liver. IGF-1 stimulates systemic body growth [20], and is an important regulator of fetal growth [21]. IGF-1 has insulin-like effects on adipocytes and lipid metabolism [22]. Cord C-peptide immunoreactivity (CPR) is a crude measure of fetal insulin secretion [23]. Hence, these plasma markers may influence fetal and placental lipid metabolism, subsequently affecting fetal fat deposition.

Based on this background, this study aimed to investigate the adipokines in cord blood that affect fetal adiposity during gestation. Considering that fetal fat mass accumulates mostly in the third trimester, we hypothesized that cord adipocytokines are associated with fetal adiposity, especially during late pregnancy.

## 2. Results

Maternal sociodemographic and clinical characteristics are shown in Table 1, and fetal ultrasonography-derived parameters are shown in Table 2. Mean (±standard deviation) cord plasma levels of leptin, adiponectin, IGF-1, and CPR were 9.1 ± 5.9 ng/mL, 23.8 ± 7.2 µg/mL, 52.8 ± 18.2 ng/mL, and 0.5 ± 0.3, respectively.

In the bivariate analyses, cord plasma leptin levels were not associated with EFA at 24 weeks but were significantly correlated with EFA at 30 weeks (r = 0.237, *p* = 0.022) and 36 weeks (r = 0.450, *p* < 0.001) (Table 3) (Figure 1). Cord plasma IGF-1 levels were not associated with EFA at 24 or 30 weeks but were significantly correlated with EFA at 36 weeks (r = 0.248, *p* = 0.016) and EFW at 36 weeks (r = 0.206, *p* = 0.047) (Figure 2).

Among potential confounders, EFA levels were significantly higher in female neonates than in males. Infant sex and gestational weight gain were associated with leptin levels, whereas infant sex was associated with both adiponectin and IGF-1 levels. (Figure 3) Gestational weight gain was associated with C-peptide. After adjusting for covariates, leptin was not associated with EFA at 24 weeks but was significantly correlated with EFA at 30 weeks (unstandardized B = 0.615 [95% CI: 0.039–1.192], *p* = 0.037) and 36 weeks (unstandardized B = 0.919 [95% CI: 0.377–1.462], *p* = 0.001) (Table 4). Leptin levels were also positively correlated with CPR (unstandardized B = 0.351 [95% CI: 0.059–0.643], *p* = 0.019). Plasma adiponectin levels were not associated with EFA during gestation. Cord plasma IGF-1 levels were not associated with EFA at 24 or 30 weeks but were significantly correlated with EFA at 36 weeks (unstandardized B = 0.007 [95% CI: 0.000–0.015], *p* = 0.045) and EFW at 36 weeks (unstandardized B = 2.575 [95% CI: 0.259–4.891], *p* = 0.030) (Table 4).

Multiple regression analysis associating cord plasma leptin, adiponectin, IGF-1 and CPR is shown in Table 5. Cord plasma leptin levels were significantly correlated with adiponectin (*p* = 0.033) and CPR levels (*p* = 0.020). CPR levels of cord blood were also significantly correlated with adiponectin levels (*p* = 0.030).

Mean and 95% confidence interval of leptin, adiponectin and IGF-1 stratified by neonatal weight categories were shown in Figure 4.

CPR levels in cord blood were also positively associated with adiponectin levels (unstandardized B = 0.011 [95% CI: 0.001–0.022], *p* = 0.028) after adjusting for covariates but were not associated with IGF-1 levels (r = 0.136, *p* = 0.190).

## 3. Discussion

### 3.1. Principal Findings

Cord plasma leptin and IGF-1 levels were significantly correlated with fetal adiposity at 36 weeks of gestation, when fetal adiposity rapidly increased. Even after controlling for confounding factors, these associations remained statistically significant.

### 3.2. Clinical Implications

To the best of our knowledge, this is the first prospective study to investigate the association between adipokine concentrations in cord blood and fetal adiposity during gestation. Cord plasma leptin and IGF-1 levels were significantly correlated with EFA in late gestation. Our findings further extend the known association between leptin and IGF-1 with fetal growth, particularly fetal fat mass, during intrauterine life [24].

Our findings indicated that cord plasma leptin levels were associated with EFA at 30 and 36 weeks, when fetal fat deposition rapidly accelerated. Cord plasma leptin is mainly derived from fetal white adipose tissue, and a previous report showed that fetal adipocyte size was a main determinant of fetal leptin levels [25,26]. The placenta is also known to produce leptin; however, only approximately 5% of placenta-derived leptin flows into the fetal circulation [26]. Hence, placental leptin is considered to have limited effect on cord plasma leptin levels. Moreover, maternal plasma leptin does not cross the placenta [27]. Based on these previous reports, cord blood leptin levels may reflect fetal fat mass [28], which is consistent with the result of the present study. On the other hand, cord plasma leptin levels were not associated with EFW, which primarily reflects skeletal growth, whereas leptin levels were strongly associated with adiposity.

Cord plasma IGF-1 levels were associated with EFA at 36 weeks. Cord plasma IGF-1 levels correlate with bone mass, lean mass, and fat mass at birth [24], which aligns with the present findings. IGF-1 stimulates adipocyte proliferation and differentiation in a dose-dependent manner [21,29]. IGF-1 does not cross the placenta, and the principal source of cord IGF-1 is the fetal liver. In sheep, fetal plasma IGF-1 levels are primarily regulated by nutrient supply and fetal insulin secretion [30]. Cord plasma IGF-1 levels were also associated with EFW at 36 weeks. Cord IGF-1 is a determinant of, or correlated with, skeletal size [24]. Nevertheless, IGF-1 showed a stronger association with EFA than with EFW at 36 weeks, consistent with prior reports indicating a particularly strong association between IGF-1 and fat mass [24].

In the present study, cord adiponectin levels were not associated with fetal adiposity. Although adiponectin is exclusively secreted by adipose tissue in adults [16,31], it is secreted not only by adipose tissue but also by muscle and vascular cells in fetal tissue, which may explain our findings. Adiponectin is especially related to visceral fat mass in adults [32]; however, we did not assess fetal visceral fat mass. Measurement of fetal visceral fat could clarify whether an association exists between visceral fat and cord blood adiponectin levels. Differentiated brown adipocytes express adiponectin genes, which are under different hormonal regulation than that observed in white adipose tissue [33]. Conversely, some reports have suggested that cord blood adiponectin concentrations are positively associated with cord blood leptin levels, fetal growth, and birth weight [34,35,36,37]. In the present study, cord plasma adiponectin levels were positively correlated with leptin levels, consistent with earlier findings. Additionally, cord adiponectin levels were significantly higher in women who delivered vaginally than in those who underwent cesarean section. Adiponectin is downregulated by glucocorticoids, which increase during vaginal delivery in vitro [38,39], supporting our observation.

### 3.3. Research Implications

Cord plasma leptin levels were also associated with cord CPR levels, a marker of fetal insulin resistance, consistent with previous reports [12]. Consequently, fetal hyperinsulinemia suppresses triglyceride degradation and promotes triglyceride synthesis [40], ultimately increasing fetal adiposity. This could represent one pathway through which cord leptin levels are linked to fetal fat accumulation.

Among adipocytokines, leptin is associated with insulin resistance via shared signaling pathways—such as JAK2/STAT3, MAPK, and PI3K—with insulin in the placenta. The insulin/IGF-1 signaling axis includes key effectors that activate STAT3 signaling. STAT3 can also be activated through alternative mechanisms, including JAK/STAT3/RACK1, AKT/STAT3, AKT/PKM2/STAT3, and AKT/mTOR/STAT3 pathways [41].

IGF-1 receptors are abundant on the surface of progenitor adipocytes [21]. Previous in vitro studies have shown that activation of these receptors promotes DNA synthesis and may facilitate preadipocyte differentiation [42]. Additionally, murine models have demonstrated that adipocyte differentiation requires supraphysiological doses of insulin but can be promoted by physiological concentrations of IGF-1 [43], suggesting that IGF-1 is a potent stimulator of adipocyte differentiation and may contribute to neonatal adipose tissue accumulation.

### 3.4. Strengths and Limitations

The primary strength of this study is its prospective longitudinal design. Multiple ultrasound datasets were obtained for each patient. Another key strength was the assessment of total fetal adiposity, proxied by measuring fat mass at three anatomical sites. One limitation was the lack of assessment of adipokine levels in the placenta; however, the evaluation of adipokine concentrations on the fetal side rather than in the placenta may have greater clinical relevance. Cord plasma adipokine levels during gestation (i.e., at 24, 30, and 36 weeks) were not measured; however, cordocentesis is clinically too invasive to be performed in utero. Because we did not evaluate the proportion of cord plasma leptin derived specifically from the fetus, it remains unclear whether cord plasma leptin has a causal effect on fetal adiposity or merely reflects fetal fat mass. Intervention studies in animal models are needed to test this hypothesis. Further investigations are warranted to confirm and replicate the findings of the present study.

## 4. Materials and Methods

### 4.1. Study Population

The study population comprised 93 mother–fetus dyads recruited from 2021 to 2024 in a prospective cohort study of biological processes in human pregnancy at Keio University. Women with uncomplicated singleton pregnancies were recruited during their first trimester of pregnancy. The exclusion criteria were preexisting major medical comorbidities (hypertension or diabetes), uterine anomalies, conditions associated with neuroendocrine and immune dysfunction (endocrine, hepatic, or renal disorders), smoking, illicit drug use, congenital malformations, and chromosomal abnormalities. The study was approved by the Institutional Review Board of Keio University School of Medicine on 27 April 2021 (no. 20,110,011), and written informed consent was obtained from all mothers.

### 4.2. Prenatal Ultrasonography

Fetal 2D ultrasonography was performed at approximately 24, 30, and 36 weeks to assess biparietal diameter, abdominal circumference, and femur length on the same day as maternal blood sampling. Estimated fetal weight (EFW) was calculated using a three-parameter model [44].

Fetal adiposity was quantified at 24, 30, and 36 weeks. The mid–upper arm, mid-thigh, and abdomen were selected for fat mass measurements because these sites can be reliably assessed using ultrasonography [5,45,46,47]. Total contours of the arm and thigh were acquired by 3D ultrasonography and then analyzed offline using appropriate software (4D View 9.0, GE Healthcare, Milwaukee, WI, USA) to determine the subcutaneous fat area on standardized cross-sectional images [47]. Briefly, the midpoints of the humerus and femur were identified, and the total cross-sectional area and lean mass area were measured as previously described. The fat area was calculated as the difference between the two measurements. The percentage of fat area was defined as the ratio of fat area to the total cross-sectional area. Fetal subcutaneous fat thickness in the abdomen was measured as the high-echoic region (2–3 cm lateral to the cord insertion) in the traditional abdominal circumference view [5].

Each measurement was performed in duplicate and averaged. All ultrasound scans were performed using the Voluson E10 (GE Healthcare, Chicago, IL, USA) with a matrix array transducer (RM6C). Fetal adiposity measurements were performed by a single obstetrician trained in fetal ultrasonography (J.T.). The intraobserver coefficients of variation for arm percent fat area, thigh percent fat area, and anterior abdominal wall thickness were 6.8%, 6.7%, and 6.0%, respectively.

### 4.3. Umbilical Cord Blood Analyses

Umbilical cord venous blood was collected at delivery, and a 9 mL blood sample was drawn by venipuncture into siliconized EDTA vacutainers and chilled to 4 °C immediately. After centrifugation of whole blood at 3000 rpm for 10 min, the plasma was decanted into polypropylene tubes, and plasma samples were frozen at −80 °C until analysis.

### 4.4. Measures of Umbilical Cord Adipokine Levels

Plasma leptin concentrations were measured by radioimmunoassay with human leptin radioimmunoassay kits (Millipore Corporation, DENIS Pharma Co., Tokyo, Japan) [27,48]. The intra-assay coefficient of variation for leptin levels was 7.71%. Plasma adiponectin concentrations were measured using a latex agglutination turbidimetric immunoassay with human adiponectin latex kits and an AND calibrator set (LSI Medience Corporation; Eiken Chemical Co., Ltd., Tokyo, Japan). The intra-assay coefficient of variation for adiponectin levels was 2.46%. IGF-1 levels were determined using an electrochemiluminescence immunoassay with Elecsys IGF-1 reagent, Eclosis Clean Cell M, and Eclosis Procell G2 Reagent (Roche Diagnostics Co., Indianapolis, IN, USA). The inter-assay coefficient of variation for IGF-1 was 2.4%. CPR levels were determined using chemiluminescent Enzyme Immunoassay (CLEIA) with C-peptide calibrators (Fujirebio Inc., Tokyo, Japan). The inter-assay coefficient of variation for CPR was 1.89%.

## 5. Data Analysis

### 5.1. Adjustment of Fetal Measures for Gestational Age at Ultrasonography

Gestational age was confirmed at the time of recruitment using an algorithm combining the last menstrual period and fetal biometry according to standard clinical criteria [49]. The gestational age of the participants varied according to the timing of the ultrasonography (three antenatal study visits ranged between 23 and 25; 28 and 31; and 35 and 37 weeks, respectively). The average gestational age at each visit (24.7, 30.5, and 36.2 weeks, respectively) was calculated to correct for the effects of variations in gestational age at assessment and to residualize the ultrasound measurements for these mean gestational ages. Briefly, the product of the regression coefficient and centered gestational age was calculated after confirming the linear relation between fetal ultrasound measures and gestational age at the time of scan, as previously described. This product was added to or subtracted from the measurement to calculate the adjusted value for the fetal parameters. This procedure standardized fetal measurements at the central gestational ages for all participants and allowed for inter-participant comparisons [5].

### 5.2. Estimated Fetal Adiposity (EFA)

EFA, composed of arm and thigh percentage fat areas and anterior abdominal wall thickness [5,50], has been presented as a relatively new parameter for quantifying fetal body composition. We have previously demonstrated the feasibility of this method in predicting the percent body fat mass of newborns [5]. EFA was calculated as the average of the standardized (z) scores of each parameter owing to the different measurement units for these three parameters.

### 5.3. Pre-Pregnancy Body Mass Index (BMI) and Gestational Weight Gain (GWG)

Maternal pre-pregnancy BMI was calculated using self-reported pre-pregnancy weight and height measured at the initial visit. Participants were classified as underweight, normal weight, overweight, or obese (BMI < 18.5, 18.5 ≤ BMI < 25, 25 ≤ BMI < 30, or BMI ≥ 30 kg/m^2^, respectively). Maternal GWG and weekly GWG (GWG/week) were also calculated. Based on the recommendations of the Japan Society of Obstetrics and Gynecology [51], optimal GWG was defined as 12–15, 10–13, 7–10, and ≤5 kg for underweight, normal weight, overweight, and obese participants, respectively. Birth weight percentiles standardized for gestational age at birth and infant sex were calculated with reference to the national database [52].

### 5.4. Neonatal Body Weight Categories

Birth weight and infant sex were obtained from medical records. Birth weight per centiles were calculated using Japanese growth charts, standardized for sex, parity, and gestational age at birth [52]. Birth weight below 10th percentile was classified as small for gestational age (SGA). A birth weight over the 90th percentile was classified as large for gestational age (LGA).

## 6. Statistical Analysis

Because of the skewed distribution of cord plasma leptin levels, a logarithmic transformation was performed to approximate a normal distribution. The association of plasma leptin, adiponectin, and IGF-1 in cord blood with EFA was examined using Pearson’s product–moment correlation. We considered the following potential confounding factors in the relation between cord plasma leptin and IGF-1 levels and fetal EFA: maternal age, parity, pre-pregnancy BMI, GWG, gestational diabetes mellitus, hypertensive disorders of pregnancy, and infant sex. Pearson’s product–moment correlations, Student’s *t*-test, and one-way analysis of variance, followed by post hoc Bonferroni testing, were used to identify confounding factors associated with either cord plasma markers (leptin, adiponectin, and IGF-1) or EFA for multivariate analysis.

Multiple linear regression was used to quantify the association between maternal plasma adipokine levels (leptin, adiponectin, and IGF-1) and EFA after adjusting for potential confounders. Additionally, the association between cord plasma leptin levels and CPR concentration was assessed using multiple linear regression analysis.

Statistical analyses were performed using IBM SPSS Statistics version 29 (SPSS Inc., Chicago, IL, USA). Statistical significance was set at *p* < 0.05.

## 7. Conclusions

Leptin in cord blood is associated with fetal adiposity at 30 and 36 weeks. Cord IGF-1 is associated with fetal adiposity and fetal weight at 36 weeks. Increased levels of leptin and IGF-1 may serve as potential plasma biomarkers of elevated fetal adiposity, which can predispose individuals to infant obesity and metabolic dysfunction later in life. To demonstrate the causal effect on leptin and IGF-1 on fetal adiposity, further investigations are warranted.

## Figures and Tables

**Figure 1 ijms-26-06926-f001:**
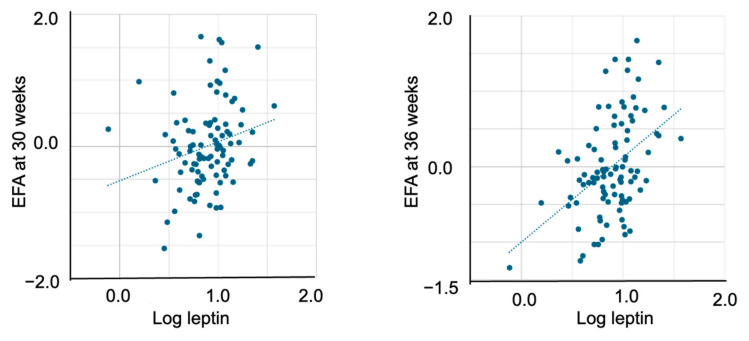
The association between cord plasma leptin and EFA at 30 and 36 weeks. Cord plasma leptin levels significantly correlated with EFA at 30 weeks (r = 0.237, *p* = 0.022) and 36 weeks (r = 0.450, *p* < 0.001). EFA, estimated fetal adiposity.

**Figure 2 ijms-26-06926-f002:**
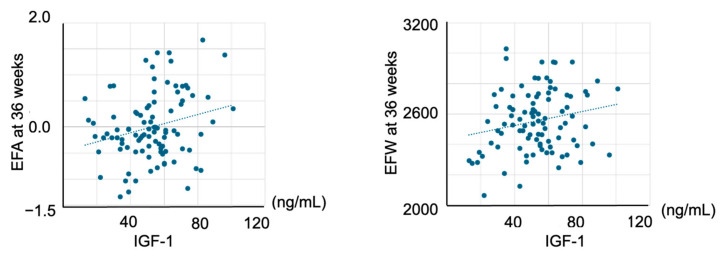
The association between cord IGF-1 levels and EFA at 36 weeks and EFW at 36 weeks. Cord plasma IGF-1 levels were significantly correlated with EFA at 36 weeks (r = 0.248, *p* = 0.016) and EFW at 36 weeks (r = 0.206, *p* = 0.047). EFA, estimated fetal adiposity; EFW, estimated fetal weight.

**Figure 3 ijms-26-06926-f003:**
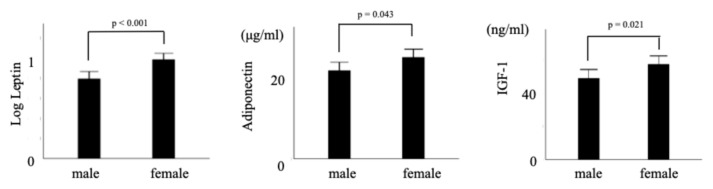
Mean and 95% confidence interval of leptin, adiponectin and IGF-1 stratified by infant sex. Cord leptin (*p* < 0.001), adiponectin (*p* = 0.043) and IGF-1 (*p* = 0.021) levels were significantly higher in female than male.

**Figure 4 ijms-26-06926-f004:**
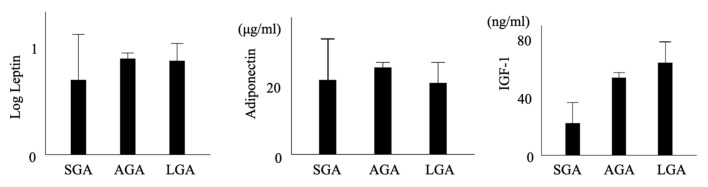
Mean and 95% confidence interval of leptin, adiponectin and IGF-1 stratified by neonatal weight categories.

**Table 1 ijms-26-06926-t001:** Maternal socio-demographic and clinical characteristics (*n* = 93).

Characteristics	
**Maternal characteristics**	
Age, years	35.0 ± 3.9
−35	54 (58.1%)
>35	39 (41.9%)
Primiparous	54 (57.4%)
Pre-pregnancy BMI, kg/m^2^	20.8 ± 2.6
Gestational weight gain, kg	13.5 ± 12.5
pre-pregnancy to 24 weeks, kg/week	0.17 ± 0.10
pre-pregnancy to 30 weeks, kg/week	0.21 ± 0.92
pre-pregnancy to 36 weeks, kg/week	0.24 ± 0.90
Gestational weight gain	
<JSOG recommendation	49 (52.1%)
=JSOG recommendation	20 (21.3%)
>JSOG recommendation	25 (26.6%)
Method of conception	
Natural Insemination	55 (58.5%)
In vitro fertilization	39 (41.5%)
Gestational diabetes mellitus	24 (25.5%)
Hypertensive disorders of pregnancy	6 (6.4%)
**Neonatal characteristics**	
Gestational age at delivery, weeks	38.8 ± 1.1
Birth weight, g	3014 ± 332
Birth weight percent tile, %	58.6 ± 26.7
SGA	4 (4.3%)
AGA	79 (84.9%)
LGA	10 (10.8%)
Infant sex (female)	47 (50.0%)

Data is shown as mean ± S.D. or *n* (%). Abbreviations: BMI, body mass index; JSOG, Japan Society of Obstetrics and Gynecology; SGA, small for gestational age; AGA, appropriate for gestational age; LGA, large for gestational age.

**Table 2 ijms-26-06926-t002:** The fetal parameters measured by fetal ultrasonography (*n* = 93).

	24 Weeks	30 Weeks	36 Weeks
Estimated fetal weight (EFW), g	757 ± 152	1578 ± 155	2553 ± 249
Mid-upper arm			
Total area, mm^2^	2.8 ± 0.5	4.7 ± 0.8	7.9 ± 1.2
Fat area, mm^2^	1.3 ± 0.4	2.4 ± 0.5	4.5 ± 1.0
Per cent fat area, %	46.8 ± 7.8	50.8 ± 6.2	56.0 ± 5.6
Mid-thigh			
Total area, mm^2^	5.7 ± 1.1	10.4 ± 1.5	17.4 ± 2.7
Fat area, mm^2^	2.2 ± 0.7	4.4 ± 0.9	8.0 ± 1.8
Per cent fat area, %	37.5 ± 5.9	41.9 ± 5.2	45.8 ± 5.1
Anterior abdominal wall thickness, mm	2.2 ± 0.5	2.9 ± 0.6	4.0 ± 0.9
Estimated fetal adiposity (EFA) *	0.01 ± 0.63	0.01 ± 0.64	0.00 ± 0.64

Data is shown as mean ± S.D. * Calculated as the average z score of arm and thigh percentage fat areas and abdominal wall thickness.

**Table 3 ijms-26-06926-t003:** Correlations between cord plasma adipokine levels and estimated fetal weight (EFW) and estimated fetal adiposity (EFA).

	EFA (24 Weeks)		EFA (30 Weeks)		EFA (36 Weeks)		EFW (24 Weeks)		EFW (30 Weeks)		EFW (36 Weeks)	
	r	*p* Value	r	*p* Value	r	*p* Value	r	*p* Value	r	*p* Value	r	*p* Value
Leptin	0.058	0.584	0.237	0.022	0.450	<0.001	−0.007	0.947	0.029	0.782	0.114	0.279
Adiponectin	−0.117	0.261	0.074	0.477	0.176	0.089	−0.018	0.865	−0.045	0.664	0.000	0.999
IGF-1	−0.070	0.505	0.141	0.176	0.248	0.016	0.188	0.069	0.127	0.223	0.206	0.047
C-peptide	−0.069	0.507	0.239	0.020	0.008	0.936	0.013	0.901	0.050	0.630	0.140	0.177

**Table 4 ijms-26-06926-t004:** Multiple regression analysis associating cord plasma metabolic biomarkers and estimated fetal adiposity (EFA) at 36 wk.

	Unstandardized Coefficients B [95% CI]	Partial Correlation	*p* Value
Leptin ^a^	1.062 [0.547–1.577]	0.399	<0.001
IGF-1 ^b^	0.007 [0.000–0.015]	0.208	0.045
adiponectin ^b^	0.013 [−0.006–0.033]	0.139	0.185
C-peptide ^c^	0.032 [−0.376–0.440]	0.032	0.875

^a^ Confounding factors: gestational weight gain, infant sex. ^b^ Confounding factor: infant sex. ^c^ Confounding factors: gestational weight gain.

**Table 5 ijms-26-06926-t005:** Multiple regression analysis associating cord plasma leptin, adiponectin, IGF-1 and CPR.

	Leptin		Adiponectin		IGF-1		C-Peptide	
	Unstandardized Coefficients B	*p* Value	Unstandardized Coefficients B	*p* Value	Unstandardized Coefficients B	*p* Value	Unstandardized Coefficients B	*p* Value
Leptin	-	-	6.216	0.033	12.808	0.103	0.343	0.020
Adiponectin	0.008	0.033	-	-	0.227	0.427	0.011	0.030
IGF-1	0.002	0.103	0.031	0.427	-	-	0.003	0.154
CPR	0.172	0.020	4.480	0.030	8.005	0.154	-	-

Confounding factors: gestational weight gain, infant sex.

## Data Availability

The data presented in this study are available on reasonable request from the corresponding author.

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
