# Peer review of "Association of Cord Blood Metabolic Biomarkers (Leptin, Adiponectin, IGF-1) with Fetal Adiposity Across Gestationâ€"

_ijms, 2025, doi:10.3390/ijms26146926_

Round 1
Reviewer 1 Report
Comments and Suggestions for Authors
Thank you for the opportunity to review this well-written, original manuscript. The study makes a valuable contribution to understanding early-life obesity determinants by exploring cord blood biomarkers and fetal adiposity. The focus on maternal-fetal metabolic health aligns with the Developmental Origins of Health and Disease (DOHaD) framework, which rightly emphasizes prenatal influences on long-term outcomes.
Major Comments
1) Title Clarification: The title suggests IGF-1 and C-peptide are adipocytokines, but they are not classified as such. Consider revising to:"Association of cord blood metabolic biomarkers (leptin, adiponectin, IGF-1) with fetal adiposity across gestation"
2) Abstract Completeness: Key results (e.g., leptin, adiponectin, EFW correlations) are missing. Please summarize all primary outcomes to reflect the full scope of findings.
3) Introduction Context: Include global prevalence data for childhood obesity/metabolic disorders to emphasize the scale and urgency of the problem (e.g., WHO/CDC statistics).
Rationale for Biomarker Selection
Justify why leptin, adiponectin, IGF-1, and C-peptide were prioritized over other biomarkers. Reference established links to fetal growth or adiposity.
4) Results Presentation. Categorize key variables:
Maternal age: Use standard groups (e.g., <20, 20–35, >35 years).
Birth weight: Classify as SGA, AGA, LGA (using INTERGROWTH-21st or WHO standards).
If maternal diet/physical activity data exist, include them as covariates or in sensitivity analyses.
5) Inconsistency in Exclusion Criteria: Table 1 reports 25.5% GDM and 6% hypertensive disorders, but the exclusion criteria (Line 193) mention preexisting comorbidities.
6) Visualization of Key Relationships: Add scatterplots or box plots showing correlations between leptin/adiponectin/IGF-1 and neonatal weight categories (SGA/AGA/LGA) or sex.
7) Discussion Gaps: Expand the discussion to include C-peptide results
8) Methods Detail: Describe C-peptide measurement methodology (e.g., assay kit, detection limits, coefficients of variation).
Incorporating these aspects could significantly enhance the originality and impact of your manuscript.
Reviewer 2 Report
Comments and Suggestions for Authors
The authors examine the association of cord blood adipokines with fetal adiposity measured by fetal ultrasonography at 24, 30 and 36 weeks gestation in 93 normal singleton pregnancies. They found cord leptin was associated with fetal adiposity at 30 and 36 weeks. Cord IGF-1 was associated with fetal adiposity and fetal weight at 36 weeks. Cord leptin and adiponectin associated with cord C-peptide. They conclude that cord leptin and IGF-1 can serve as biomarkers of fetal adiposity.
Although the sample size is not large, the use of fetal ultrasonography to estimate fetal adiposity at different gestation weeks makes the study unique. However, it is unclear what the clinical utility of cord adipokines in serving as a biomarker for fetal adiposity is.
Although the writing is clear, the tables of results can be presented more clearly. It would be helpful if the multivariate regression analyses of the 4 cord biomarkers leptin, adiponectin, IGF-1 and C-peptide with EFA and EFW at the different gestation weeks are presented together in a table as in table 3 (correlation analysis).
It would be clearer if the legend of the table could state clearly which are the confounders that were adjusted for in the multivariate analyses. In the statistical analyses section of the methods it was stated maternal age, ppBMI, parity and GDM were confounders adjusted for, but in table 4a only GWG and sex were presented.
Another table should show the correlation and association (after adjusting for covariates) between leptin, adiponectin, IGF-1 and CPR.
The abstract stated that "The multivariate analyses indicated that leptin was not associated with EFA at 24 and 30 weeks but was significantly correlated with EFA at 36 weeks". However in the results it is stated "After adjusting for covariates, leptin was not associated with EFA at 24 weeks but was significantly correlated with EFA at 30 weeks (unstandardized B = 0.615 [95% CI: 0.039–1.192], p = 0.037) and 36 weeks (unstandardized B = 0.919 [95% CI: 0.377–1.462], p = 0.001) (Table 4)."
The method for C-peptide measurement is missing, and also the units for CPR (line 56).
The blood samples appeared to be collected in EDTA blood tubes so plasma were used for analyses, but serum is mentioned in the methods and also in the abstract.
Reviewer 3 Report
Comments and Suggestions for Authors
This is an interesting research article with high novelty. Some points should be addressed.
- The authors could avoid to report the non significant results in the Abstract.
- A bit more analysis of the introductory sentences in the Abstract could be useful for the readers.
- The conclusions of the study could be a bot more analysed at the end of the Abstract.
- After the 1st paragraph of the Introduction, the authors should analyse whic maternal risk factors (e.g. excessive maternal weight gain during pregnancy, pre-pregnancy maternal obesity, etc.) may increased fetal adiposity, including relevanreferences.
- In the Introduction, the authors should describe with more details the potential role of leptin, adiponectin, and insulin-like growth factor-1 as well as C-peptide in fetal adiposity.
- The sentence in lines 118-119 needs at least a relevant reference.
- The sentence in lines 126-128 needs at least a relevant reference.
- Conclusion is too small and should be enriched by the most significant results of the study. Morever, the authors should add their opinion based on their results concerning what future studies could be performed.
Round 2
Reviewer 1 Report
Comments and Suggestions for Authors
The authors have responded well to the reviewers' comments.
I have no further concerns.
Reviewer 2 Report
Comments and Suggestions for Authors
The revised version has improved and has addressed the concerns in the previous review.
A minor point is some of the fonts are different.
Reviewer 3 Report
Comments and Suggestions for Authors
The authors have considerably improved their manuscript.